

# Seasonal and annual variability of methane emissions to the atmosphere from the surface of a eutrophic lake located in the temperate zone (Lake Kortowskie, Poland)

Andrzej Skwierawski

Department of Water Resources and Climatology, University of Warmia and Mazury in Olsztyn, Plac Łódzki 2, 10-719 Olsztyn, Poland

*Correspondence to*: Andrzej Skwierawski (andrzej.skwierawski@uwm.edu.pl)

**Abstract**

Despite studies on methane emissions from lakes, there remains considerable uncertainty in accurately estimating global
emissions from this source. This uncertainty is related to the diversity of lake types, their conditions, geographical locations, as well as the various research methods employed, typically short measurement series, temporal variability of emissions, and the various forms of emissions: diffusion, ebullition, transport in macrophytes, and storage emission.

In this study, an attempt was made to supplement information on methane emissions through real-time *in situ* measurements using a measurement chamber connected to a mobile CRDS spectrometer. Measurements were conducted on
Lake Kortowskie, which is representative of highly eutrophic lakes in the northeastern region of Poland. The methane emission measurement cycle was carried out over a full four-year period (2019-2022) at weekly intervals, alongside simultaneous observations of water indicators and meteorological measurements.

The average methane emission from the surface of Lake Kortowskie over the entire observation period was 11.79 mg m$^{-2}$ d$^{-1}$, with a median of 6.91 mg m$^{-2}$ d$^{-1}$, and a maximum of 134.4 mg m$^{-2}$ d$^{-1}$ on a single measurement date. During the four-year
observation period, slight differences in annual averages were noted, along with significant variability in seasonal emissions. In the years 2019, 2020, 2021, and 2022, the average CH$_4$ emissions were 13.7, 10.1, 11.8, and 11.6 mg m$^{-2}$ d$^{-1}$, respectively. Seasonally, average emissions were recorded at 3.2, 12.1, 20.6, and 14.9 mg m$^{-2}$ d$^{-1}$ for winter, spring, summer, and autumn, respectively.

The studies indicated that the main environmental factors associated with methane emissions from the lake were primarily
water temperature and air temperature. However, water waves height, wind speed and gusts, precipitation totals, Secchi depth, and oxygen concentration in the water also played significant roles. Regression analyses for Lake Kortowskie suggest that only changes in the main climate components, following the current trend of changes, could lead to an increase in methane emissions from the lake by over 30% by the year 2100.



## 1 Introduction

Methane is the second most important greenhouse gas after $CO_2$, significantly affecting the planetary radiative balance and contributing to global warming (Saunois et al., 2020). Since the pre-industrial era, the concentration of methane in the atmosphere has increased from approximately 700 ppb to nearly 2000 ppb (Dlugokencky et al., 2011; Mitchell et al., 2013; Oh et al., 2022), resulting in an increase in radiative forcing of 0.56 W m$^{-2}$, thus accounting for about 20% of the total anthropogenic impact on the Earth's radiative balance (Forster et al., 2023). Although the concentration of methane in the

atmosphere is significantly lower than that of $CO_2$, its greater efficiency in absorbing infrared radiation makes it a gas 25 times more potent than $CO_2$. On the other hand, the residence time of $CH_4$ molecules in the atmosphere is relatively short, averaging 10 years, making its sustained concentration a result of the balance between the rate of decay and the magnitude of emissions. Methane emissions originate from both natural and anthropogenic sources. However, there are still significant uncertainties in assessing methane fluxes within the Earth's geoecosystem (Kirschke et al., 2013; Del Sontro et al., 2016). Between 2008-2017,

natural sources accounted for approximately 40% of global emissions, i.e., 217 Tg $CH_4$ y$^{-1}$ (Saunois et al., 2020). The literature suggests that emissions from natural sources may increase with ongoing climate warming through a positive feedback mechanism (Dean et al., 2018). Additionally, the observed accelerated increase in atmospheric concentration in recent years is attributed to increased emissions from biogenic sources, as confirmed by $^{13}$C-$CH_4$ isotope studies (Schaefer et al., 2016; Lan et al., 2021; Oh et al., 2022; Zhang et al., 2023).

Lakes make a significant contribution to global methane emissions from natural sources (Johnson et al., 2022). However, the magnitude of these emissions is difficult to estimate on a global scale due to temporal and spatial variability, which depends on climate zone, lake surface area, and condition. According to Bastviken et al. (2011), global methane emissions from inland surface waters may average 103 Tg $CH_4$ y$^{-1}$, which, when converted to $CO_2$ equivalents, corresponds to about 25% of the carbon absorbed by terrestrial ecosystems. Methane emissions from lakes are estimated within a wide range. Bastviken et al.

(2004) estimate these emissions from 8 to 48 Tg $CH_4$ y$^{-1}$ (from 6 to 16% of global natural $CH_4$ emissions).

     Lakes can emit methane into the atmosphere in several ways: through the diffusion of dissolved methane from the water-air interface, ebullition in the form of bubbles of varying sizes emerging from the sediment, and transport through the tissues of aquatic plants (Bastviken et al., 2023; Van den Berg et al., 2024). Dissolved methane in the water that does not undergo oxidation processes is emitted into the atmosphere through diffusion. The intensity of this process is closely related to the

current concentration of $CH_4$ in the surface water layer and depends on turbulence and wind-induced mixing (Bastviken et al., 2004). Most lake waters are supersaturated with methane relative to equilibrium with atmospheric air, initiating the diffusion process and the release of $CH_4$ into the atmosphere (Rasilo et al., 2015; Holgerson and Raymond, 2016).

     Methane emissions from lakes have been studied since the 1980s; however, there is significant temporal and spatial variability, different emission pathways (diffusion, ebullition, and transport in macrophytes), and differences in research

methods. These factors result in high uncertainty in the obtained results, limiting the ability to accurately assess emissions from inland surface waters on a global scale (Sanches et al., 2019). Methane emissions from surface waters and wetlands are



considered a major source of uncertainty in assessing the global methane budget (Ortiz-Llorente and Alvarez-Cobelas, 2012; Kirschke et al., 2013; Del Sontro et al., 2016; Saunois et al., 2020). According to the authors of a new study assessing global methane emissions from lakes (Johnson et al., 2022), there are still issues with adequate measurement datasets, which are often
short-term, do not account for the seasonality of emissions and other lake indicators, and lack sufficient characterization of the state of the analyzed lakes and the climatic conditions of the study area.

The main objective of this study was to attempt to assess the magnitude of methane emissions from the surface of a highly eutrophic lake characterized by a moderate ecological state, like most lakes in Poland. The research aimed to track the variability of emissions on a seasonal basis over four years of regular observations (2019-2022), in relation to meteorological
conditions and water quality indicators of the lake. The study was conducted using the method of direct real-time *in-situ* emission measurements, using CRDS (cavity ring-down spectroscopy) method and a closed flow chamber. Measurements were carried out in the northeastern region of Poland (Masurian Lake District), where methane emission studies from lakes likely have not been conducted previously (or at least the author did not find such publications). An attempt was also made to create a simple model to predict possible future changes in $CH_4$ emissions from the studied lake.

## 2 Materials and methods

### 2.1 Study site characteristics

The research focused on Lake Kortowskie, located in a temperate climate zone in northeastern Poland, in the region of the Masurian Lake District (mesoregion of Olsztyn Lakeland). The research site is located at 53.761°N 20.445°E. The lake has an area of 91.8 hectares (measured based on orthophotomaps and digital terrain models), a maximum depth of 17.2 meters, an
average depth of 5.9 meters, and a volume of 5.3 million cubic meters (Dunalska et al., 2007). The lake is situated within the city of Olsztyn, partially within the campus area of the University of Warmia and Mazury. The total catchment area of the lake covers 3,860 hectares, with approximately 50% covered by forests, 12% consisting of surface waters (including Lake Ukiel with an area of 412 hectares), and 10% being urban areas (Fig. 1). Lake Kortowskie has five small surface inflows, and the water turnover rate is approximately 70% per year. In the past (until 1990), the lake received municipal sewage (Augustyniak-
Tunowska et al., 2023).

Since 1956, Lake Kortowskie has been the subject of experimental restoration using a method called selective hypolimnetic withdrawal, which is likely the longest-running lake restoration experiment in the world. The system operated at full capacity (0.25 $m^3$ $s^{-1}$) during the years 1976-1989 and 1995-1998 during summer and winter stagnation periods (Dunalska et al., 2007). In later years, it was operated only in the summer, with a reduced capacity of 0.08 $m^3$ $s^{-1}$ (Bowszys and Bogacka-
Kapusta, 2021). During the period of the present study (2019-2022), the experiment was not in operation.

Limnologically, the lake represents a dimictic type, with an epilimnion extending to a depth of approximately 4 meters, a less sharp metalimnion, and a hypolimnion with a temperature of around 8°C (Fig. 1). Similar profiles for Lake Kortowskie



have been obtained since the 1950s, although during periods when the hypolimnetic water withdrawal system operated at full capacity, thermal stratification patterns in the southern part of the lake were partially disrupted (Dunalska et al., 2007).


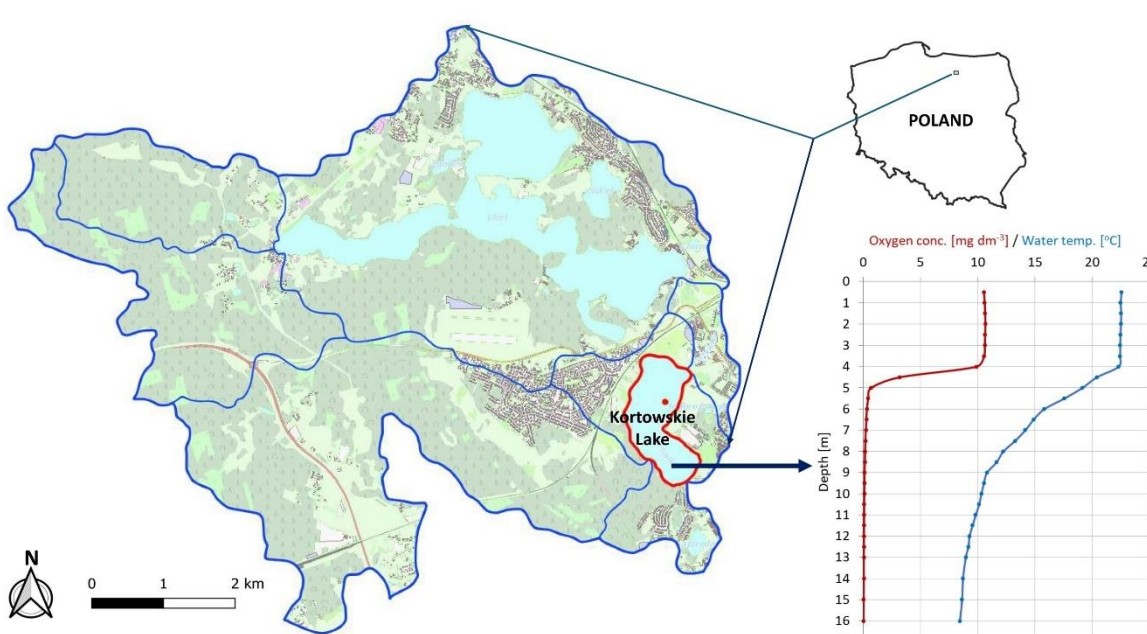

**Figure 1.** The location of Lake Kortowskie catchment area (divided into partial catchments by the blue line), the contour of the lake shoreline (red line), and the thermal and oxygen profile of the lake, obtained at the deepest point of the lake in August 2022; the catchment map base is derived from the database of topographic objects (BDOT) and used under the terms of the open resource of the geoportal.gov.pl website.


According to research conducted as part of the national water monitoring in Poland, related to the European Union's Water Framework Directive, Lake Kortowskie was monitored four times between 2015 and 2021, consistently achieving a moderate ecological status, classified as class 3/5 (data from the Chief Inspectorate of Environmental Protection of Poland, https://wody.gios.gov.pl/pjwp/publication/LAKES/87), failing to meet the requirements for good ecological status due to the
condition of biological elements (phytoplankton, fish fauna) and additional elements (oxygen saturation of hypolimnetic water, phosphorus concentration). Lakes with a moderate ecological status accounted for 52% of the lakes surveyed as part of the monitoring related to the Water Framework Directive in Poland between 2016 and 2021 (Statistics Poland, 2022), thus, the studied lake can be considered a typical representative of highly eutrophic lakes in the region.

The period of the study was characterized by higher annual average air temperatures compared to the long-term average
(Table 1), with the year 2021 being close to the average for the period 1991-2020, while the year 2019 was the warmest year in Olsztyn since 1951. The summer season, which is most significant in terms of methane emissions, was also noticeably warmer than the long-term average in all study years. Winter seasons during the study period were mild, except for a cold



period at the beginning of 2021 (Fig. 2). Additionally, atmospheric precipitation and wind conditions were similar to the long-term average values, although the year 2022 was characterized by dry conditions.


**Table 1.** Climatic conditions in Olsztyn in the years of observation (2019-2022) compared to the averages from the reference periods 1951-1980 and 1991-2020.

| Parameter | Period/Year | | | | | |
|---|---|---|---|---|---|---|
| | 1951-1980 | 1991-2020 | 2019 | 2020 | 2021 | 2022 |
| Avg. annual temperature [°C] | 6.8 | 8.0 | 9.6 | 9.5 | 8.1 | 8.9 |
| Seasonal avg. temperature [°C]: | | | | | | |
| Winter | -2,8 | -1,2 | 0,9 | 2,4 | -2,3 | 0,9 |
| Spring | 6,0 | 7,6 | 8,7 | 7,2 | 6,7 | 7,1 |
| Summer | 16.5 | 17.4 | 19.0 | 18.2 | 19.1 | 18.9 |
| Autumn | 7,6 | 8,2 | 9,7 | 10,3 | 8,9 | 8,8 |
| Days with max temperature >25°C | 26.5 | 35.6 | 50 | 38 | 41 | 44 |
| Annual precipitation sum [mm] | 628.3 | 642.4 | 672.2 | 667.5 | 677.3 | 486.9 |
| Avg. relative humidity [%] | 81.8 | 79.9 | 76.3 | 77.8 | 79.5 | 77.4 |
| Avg wind speed [m s$^{-1}$] | 3.2 | 2.9 | 3.0 | 3.0 | 2.9 | 2.8 |

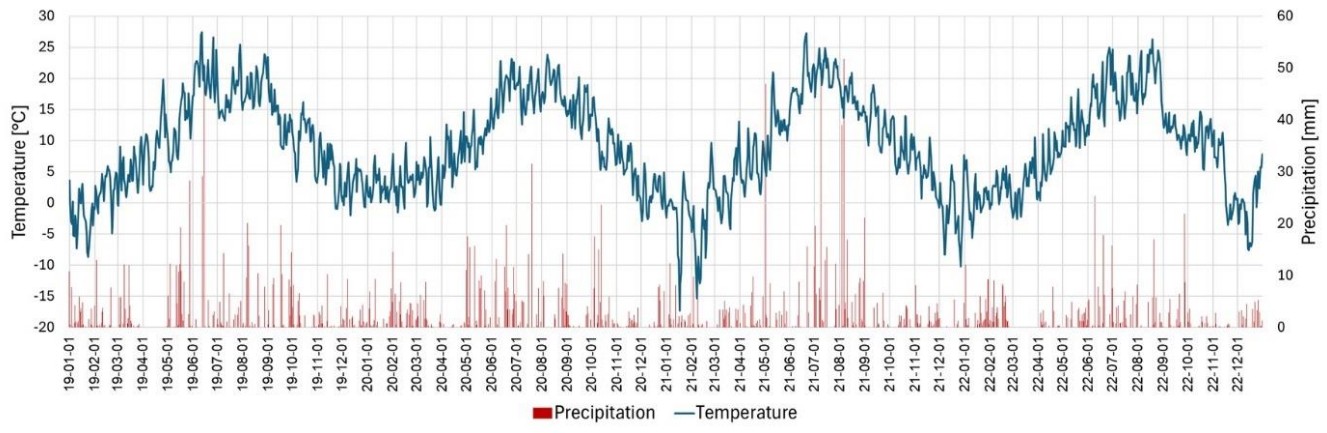


**Figure 2.** The course of changes in the average daily air temperature and daily precipitation sums in Olsztyn from 2019 to 2022.



**2.2 Methods**

Measurements of methane emissions were conducted *in situ* in real-time using a mobile cavity ring-down spectroscopy (CRDS) spectrometer, model GasScouter G4301 (Picarro, Santa Clara, CA, USA), equipped with a chamber for gas exchange measurements on soil and water surfaces. The research was carried out throughout the entire four-year period from 2019 to 2022, at approximately weekly intervals. A total of 198 measurement cycles were conducted, during which a total of 990 3-

minute emission measurements were taken, with 11 of these measurements being discarded during data processing due to being deemed faulty. The representativeness of the measurement period was verified by calculating the correlation between the monthly mean air temperature values at the time of measurement and the standardly calculated monthly mean air temperatures from the meteorological dataset. The correlation between these datasets was 0.976 (p<0.001). Correlations between the mean wind speed at the time of measurement for a given month and the mean wind speed for the month also

showed statistically significant relationships, although due to the high short-term variability of wind speed, this relationship was less precise, with a correlation coefficient of 0.295 (at the statistical significance threshold of p=0.04).

Measurements were conducted at 5 designated measurement sites, located from the boundary of the littoral zone adjacent to the reed belt (depth of 1.5 m) to the pelagic zone within the epilimnion range (depth of 3.5 m). The sites were determined using three permanent piers, the surroundings of which are not used as bathing areas (bathing is prohibited), ensuring that the

sediment at these locations is not disturbed by direct anthropogenic pressure. Measurements were taken during midday hours (from 11:00 to 14:00), for approximately 4 minutes at each site, using the GasScouter device, which sampled air in a closed chamber at an average frequency of 72 readings per minute. To calculate the current emission rate based on the obtained methane concentration curve in the chamber, precisely 180 seconds of stable methane concentration change were selected in the measurement system. The unit emission rate (eCH$_4$, mg m$^{-2}$ d$^{-1}$) was calculated using the formula:


$$eCH_4 = \frac{(C_{180} - C_0) \cdot \rho \cdot 0.004 \cdot 480}{0.0615}$$

where:

C$_0$ is methane concentration in the air in the measurement system at the beginning of the measurement [ppmv];

C$_{180}$ is methane concentration in the air in the measurement system after 180 seconds of measurement [ppmv];

ρ is the volumetric density of methane [kg m$^{-3}$], based on physical tables, individually determined for the temperature of each measurement (t, °C), according to the formula:

$$\rho = 0.000009 \cdot t^2 - 0.0026 \cdot t + 0.7083$$


0.004 is the volume of the closed measurement system [m$^3$];



480 is the time conversion factor from 3 minutes to a day;

0.0615 is the surface area of the water surface covered by the measurement [$m^2$].

To obtain more reliable mean values from each measurement session, two extreme values (maximum and minimum) out of the five values obtained at each measurement site were discarded, and the remaining three results were used to calculate the mean emission value for that session. This procedure reduced the average relative standard deviation between measurements from 45% to 26% across the entire dataset while minimally affecting the mean emission value for the entire study period. The mean emission value from all measurements over the study period was 12.36 mg $m^{-2}$ $d^{-1}$ with a standard deviation (SD) of

5.78. After discarding two measurements, the emission was 11.79 mg $m^{-2}$ $d^{-1}$ with an SD of 3.44. That method of excluding extreme values was employed to increase the data reliability, especially during measurement sessions under challenging weather conditions (strong winds, water waving, high temperatures). A short-term measurement (3 minutes) using the chamber created a measurement environment independent of the influence of conditions that could interfere with the measurement: 1) changes in conditions within the chamber (temperature, water vapor saturation, increasing $CH_4$ concentration in the air); 2)

minimizing the impact of wave damping by positioning the chamber on the water surface.

In practice, the applied method measured the diffusive emission of methane from the open water surface. However, ebullition involving large bubbles occurred only occasionally during the entire study period and only at the shallowest (1.5 m) research site. Among the 979 sampling events, 12 such measurements (1.2% of all measurements) were recorded. However, due to the inability to accurately balance these events, such measurements were deemed unsuccessful and the measurement at

the site was repeated. The water surface area covered by the chamber was small (0.0615 $m^2$), but considering the approximately 58 hours of total net measurement time, these results suggest a relatively minor significance of ebullition in the studied lake. Water physicochemical indicators in the lake were measured concurrently with methane emission measurements using a YSI6600 multiparameter probe (YSI Incorporated, Yellow Springs, OH, USA) (observations in 2019-2020), which was replaced by a newer model, YSI EXO2, in subsequent years (2021-2022). Measurements were taken along the epilimnion

profile at approximately 0.1 m vertical intervals to a depth of 3.5 m, covering indicators such as temperature [°C], electrolytical conductivity [μS $cm^{-1}$], dissolved oxygen concentration [mg $dm^{-3}$], oxygen saturation [%], pH, redox potential [mV], chlorophyll-a concentration [μg $dm^{-3}$], and turbidity {NTU}. From these measurements, the average value for the upper 1 m layer of water was calculated for further analysis. Table 2 presents the average values and range for these indicators over the study period, as well as their seasonal averages. The temperature gradient in the epilimnion was determined as the difference

between the average temperature of the upper 1 m and the average value from depths of 3-3.5 m.

Additionally, the Secchi depth [m] was measured each time using a standard 20-cm black-and-white disk with a marked rope. Water waving, as the average wave height, was also measured using a measuring stick at the end of the pier over the lake. Due to the difficulty of precisely determining wave phenomena (due to high variability), a simplified scale based on the Beaufort scale was used for further analysis. The scale was doubled and applied in intervals of 0.5 degrees: 0 (no waves); 1:

average wave height of 5 cm; 2: 10 cm; 3: 15 cm; 4: 20 cm; 5: 25 cm. No waves of degree 5 occurred during the observation



period. The concentration of methane in the atmosphere was also measured, conducting a 10-15 minute measurement over the lake surface during each measurement session, using GasScouter spectrometer.

**Table 2.** The average and seasonal values of basic water quality indicators in the surface layer (0-1 m) of Lake Kortowskie from 2019 to 2022.

| Parameter | 2019-2022 | | Seasonal averages (2019-2022) | | | |
|---|---|---|---|---|---|---|
| | Average | Range | Winter | Spring | Summer | Autumn |
| Temperature [°C] | 11.68 | 0.72÷27.79 | 3.11 | 11.47 | 22.38 | 12.52 |
| Oxygen saturation [%] | 101.6 | 51.9÷196.9 | 90.6 | 110.2 | 122.2 | 83.1 |
| Oxygen conc. [mg/l] | 11.12 | 5.95÷24.30 | 12.15 | 12.14 | 10.58 | 8.83 |
| Conductivity [µS/cm] | 406.6 | 332.5÷442.0 | 421.8 | 418.4 | 387.9 | 392.4 |
| pH range | - | 7.72÷9.53 | 7.72÷9.39 | 8.27÷9.31 | 8.00÷9.53 | 8.11÷9.02 |
| Redox potential [mV] | 132.9 | 37.1÷260.9 | 132.5 | 142.1 | 130.5 | 127.5 |
| Chlorophyll-a [µg/l] | 10.10 | 0.72÷43.1 | 9.10 | 12.25 | 7.03 | 14.10 |
| Turbidity [NTU] | 2.28 | 0.47÷10.45 | 1.30 | 1.75 | 3.60 | 2.33 |
| Secchi depth [m] | 1.93 | 0.70÷3.60 | 2.59 | 1.85 | 1.49 | 1.41 |

Meteorological data were obtained from the repository of state data, the measurement network of the Institute of Meteorology and Water Management, National Research Institute (IMGW-PIB), for the Olsztyn station located in close proximity, about 1.5 km from the research site. This is a synoptic station that records meteorological measurements on an hourly basis. For the purposes of this study, current meteorological data from the time of measurement were used, and daily and monthly data were developed for basic meteorological indicators: temperature [°C], dew point temperature [°C], wind speed and gusts [m s$^{-1}$], relative humidity [%], cloud cover [8-point scale], atmospheric pressure [hPa]. Data from multi-year observation periods since 1951 were also used. The seasons for data analysis purposes were determined based on thermal criteria (average daily temperature), divided into: winter < 5°C, spring and autumn 5-15°C, summer >15°C. In the case of using monthly data, the seasons were divided according to meteorological criteria: winter is December-February; spring is March-May; summer is June-August, and autumn is September-November.

The obtained results were statistically analyzed using Statistica 13 software from Statsoft Poland, employing basic statistics, Spearman's correlation analysis, regression analysis, and principal component analysis (PCA) for data from individual measurements (n = 198) and data averaged on a monthly basis (n = 48).





## 3 Results and discussion

### 3.1 Annual and seasonal methane emissions

Throughout the four-year observation period (2019-2022), the average methane emission from the surface of Lake Kortowskie was 11.79 mg m$^{-2}$ d$^{-1}$, with a median value 6.91 mg m$^{-2}$ d$^{-1}$. At individual measurement sites, the average results slightly differed: from 9.5 mg m$^{-2}$ d$^{-1}$ at the shallowest site (just behind the littoral vegetation belt, depth of 1.5 m), 11.6 mg m$^{-2}$ d$^{-1}$ on average at two farther sites (depth of 2.5 m), to 14.6 m$^{-2}$ d$^{-1}$ at two most exposed points in the open water zone (depth of 3.5 m). During the 4-year observation period, relatively small differences in averages were noted in individual years, along with significant seasonal variability in emissions. In the years 2019, 2020, 2021, and 2022, the average $CH_4$ emissions were as follows: 13.7, 10.1, 11.8, and 11.6 m$^{-2}$ d$^{-1}$, respectively. Over the seasonal cycle of the entire 4-year study period, average emissions were observed at the following levels: 3.2, 12.1, 20.6, and 14.9 m$^{-2}$ d$^{-1}$ for winter, spring, summer, and autumn, respectively. However, values recorded in successive measurements showed significant variation, and extreme values obtained deviated considerably from the averages (Fig. 3).

The recorded annual emission, averaged over the four years of study, was relatively high but did not exceed the global average for lakes. According to a global estimation prepared by Johnson et al. (2022), methane emissions from lakes via diffusion amount to 14.1 Tg $CH_4$ y$^{-1}$, which, when converted to unit area, yields a result of 13.8 mg m$^{-2}$ d$^{-1}$. An earlier estimate for inland waters in the geographic zone of 25-54° latitude (Bastviken et al., 2011) provides a value of 4.8 Tg $CH_4$ y$^{-1}$, which, considering the total lake area, gives an average unit emission of 9.9 mg m$^{-2}$ d$^{-1}$. However, Lake Kortowskie is located at the northern edge of this zone (53.76° N), where methane emission values are typically much lower than in areas of lower latitudes. For waters within the latitude range of 54-66°, the estimated emission from a similar total area resulted in 1.1 Tg, indicating an average emission of 1.97 mg m$^{-2}$ d$^{-1}$. Lake Kortowskie yielded a significantly higher result compared to these values, indicating a relatively high methane emission in its category. The average emission value from Lake Kortowskie was very close to the average emissions from coastal lakes of the Baltic Sea in Poland, which amounted to 12.7 mg m$^{-2}$ d$^{-1}$ (Waszczyk and Schubert, 2020), although coastal lakes have a larger surface area, are shallower, and are much more exposed to wind. The average emission from Lake Kortowskie also exceeded the values of each lake in the USA and Sweden, which were studied by Bastviken et al. (2004). According to Holgerson and Raymond (2016), average methane emissions are inversely proportional to lake surface area (smaller lakes emit more). According to their estimation, lakes with a surface area of 10-100 ha emit an average of 3.4 mg m$^{-2}$ d$^{-1}$ indicating that the values observed in Lake Kortowskie were over 3 times higher than the average in this size class.





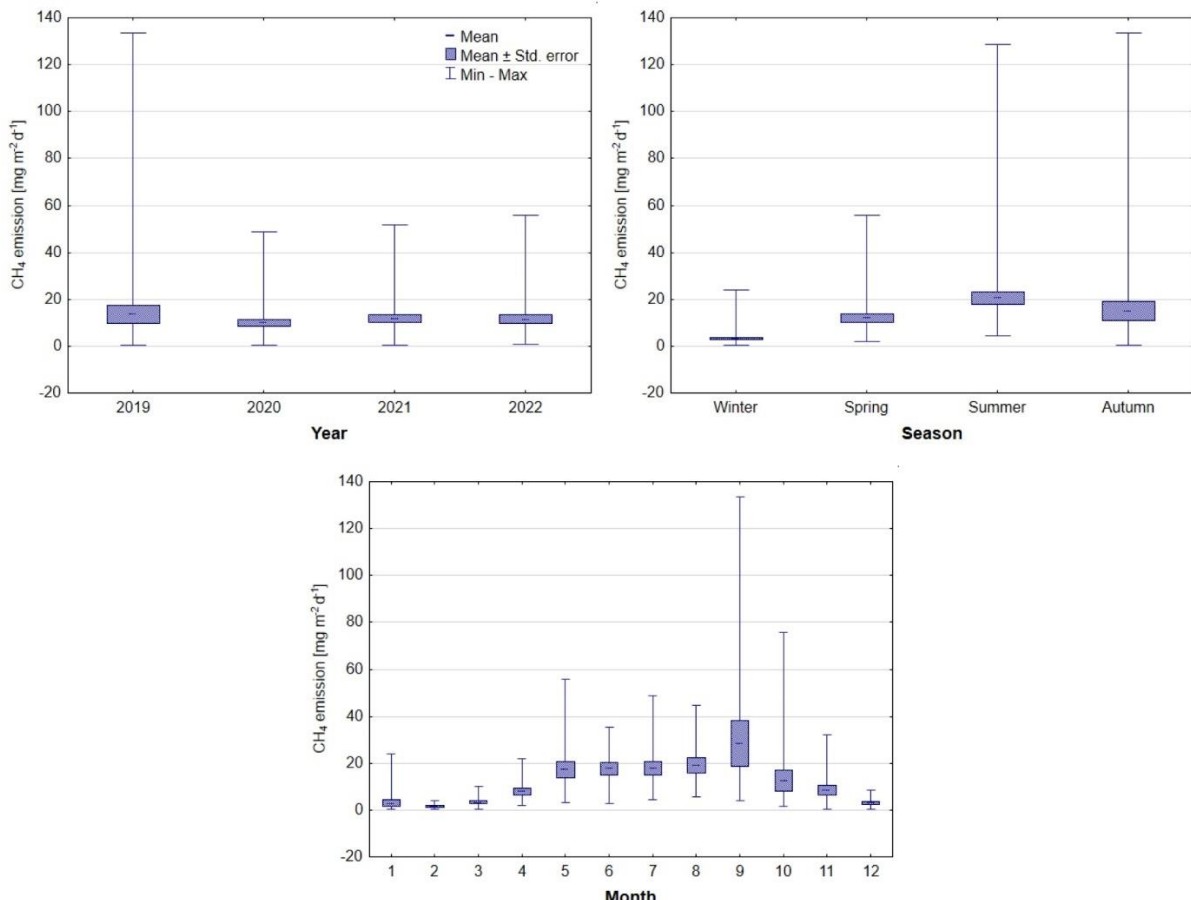


**Figure 3.** Variability of CH$_4$ emissions from the surface of Lake Kortowskie in individual years, seasons, and months throughout the entire study period (2019-2022): average values, standard error of the mean, as well as minimum and maximum extreme values.

It is worth emphasizing that all measurements taken (n=979) yielded positive values for CH$_4$ emissions, indicating that
the lake under study was consistently supersaturated with this gas compared to the equilibrium state with methane present in the atmospheric air above the water surface. The lowest recorded emission value was 0.27 mg m$^{-2}$ d$^{-1}$, with all results below 1 mg m$^{-2}$ d$^{-1}$ (n=22) noted between December and early March, typically under conditions of water calmness (no waves). Negative CH$_4$ emissions (uptake from the atmosphere) are very rarely observed in studies of water bodies. Demarty et al. (2009) obtained negative CH$_4$ emissions (minimum value -0.19 mg m$^{-2}$ d$^{-1}$) in investigations of retention reservoirs in Canada,
obtained through automatic measurements based on the partial pressure of pCH$_4$ and gas exchange at the water-air interface, with average emissions from these facilities typically < 1 mg m$^{-2}$ d$^{-1}$. Similarly, Suomis et al. (2004) reported negative minimum values of CH$_4$ emissions, at -1.5 mg m$^{-2}$ d$^{-1}$ in one of the retention reservoirs in the western United States. Very low CH$_4$ emissions were recorded in retention reservoirs in the subalpine region, averaging only 0.2 mg m$^{-2}$ d$^{-1}$, with the highest recorded value at 1.8 mg m$^{-2}$ d$^{-1}$ (Diem et al., 2012).



Periods of typical limnological winter on the studied lake, i.e., longer presence of ice cover across the entire lake, occurred only twice during the study period: from mid-January to mid-February 2019 and from mid-January to mid-March 2021. During these periods (n=14), methane emissions were measured as "potential" meaning measurements were taken in holes drilled in the ice with a diameter larger than that of the measurement chamber. The average $CH_4$ emission value from these periods was 3.4 mg m$^{-2}$ d$^{-1}$, which was similar to the average of all winter measurements, amounting to 3.2 mg m$^{-2}$ d$^{-1}$ (n=70), although one

measurement from January 22, 2021, significantly influenced the result, with an average value of 24.0 mg m$^{-2}$ d$^{-1}$, where high emission values were recorded at 3 stations, and the course of changes in $CH_4$ concentration in the chamber during measurement indicated that it was not ebullition phenomenon. The potential emissions from the ice-covered periods accounted for a total of 2.05% of the overall emissions for the entire study period, with the time share of this period being 4.7%.

     In the monthly analysis across individual years, the seasonal variation in $CH_4$ emissions is particularly evident, as observed

throughout the entire study period (Fig. 3). Significantly higher emissions were observed during the summer and autumn months, while low emissions occurred during the winter months and early spring (December-March), except for the year 2021, when elevated $CH_4$ emission values were recorded in January. A gradual increase in emissions during the spring months is also clearly visible (Fig. 4). However, there was some variability in different years, which can be attributed to the variability in weather conditions (Fig. 2).

Very high emissions were recorded in September 2019, occurring in two consecutive measurements in the middle of that month (September 11 and 16, 2019). The highest recorded average emission values in the entire dataset (n=198) were obtained during these periods, amounting to 128.8 and 134.4 mg m$^{-2}$ d$^{-1}$. The highest single measurement result recorded during these days was 264.4 mg m$^{-2}$ d$^{-1}$. Extremely high emission results coincided with a period of decreasing average daily air temperature, which dropped from 23.7°C on September 1 to below 10°C on September 17. The water temperature also decreased by

approximately 5°C during this time. These two episodes of extremely high emissions also coincided with increased wind speed (especially in gusts), strong water waving, a decrease in water oxygen saturation, as well as a decrease in pH and redox potential values. All of this may indicate an event involving the "flushing" of the upper metalimnion surface, from which deoxygenated water (potentially rich in methane) was mixed with epilimnion water. However, this was not yet the beginning of autumnal circulation, as the water temperature on September 16 was at 17.5°C. It can be assumed that in September 2019, such a

phenomenon occurred suddenly, causing additional methane emissions, referred to in the literature as "storage emissions" (Fernandez et al., 2014; Li and Xue, 2021), which are a particular case of diffusion (Bastviken, 2009). Storage emissions result in elevated methane emissions to the atmosphere, which are recorded during periods after the spring ice cover melts and during full mixing in autumn, after the disappearance of summer thermal stratification (Riera et al., 1999; Kankaala et al., 2006). Such emissions result from methane storage in deoxygenated hypolimnion during summer stratification. In Fernandez et al. (2014)

research, it was found that on average, 46% of the methane stored in the hypolimnion was emitted into the atmosphere during autumnal circulation, and the impact of these emissions accounted for 80% of the total lake emissions.



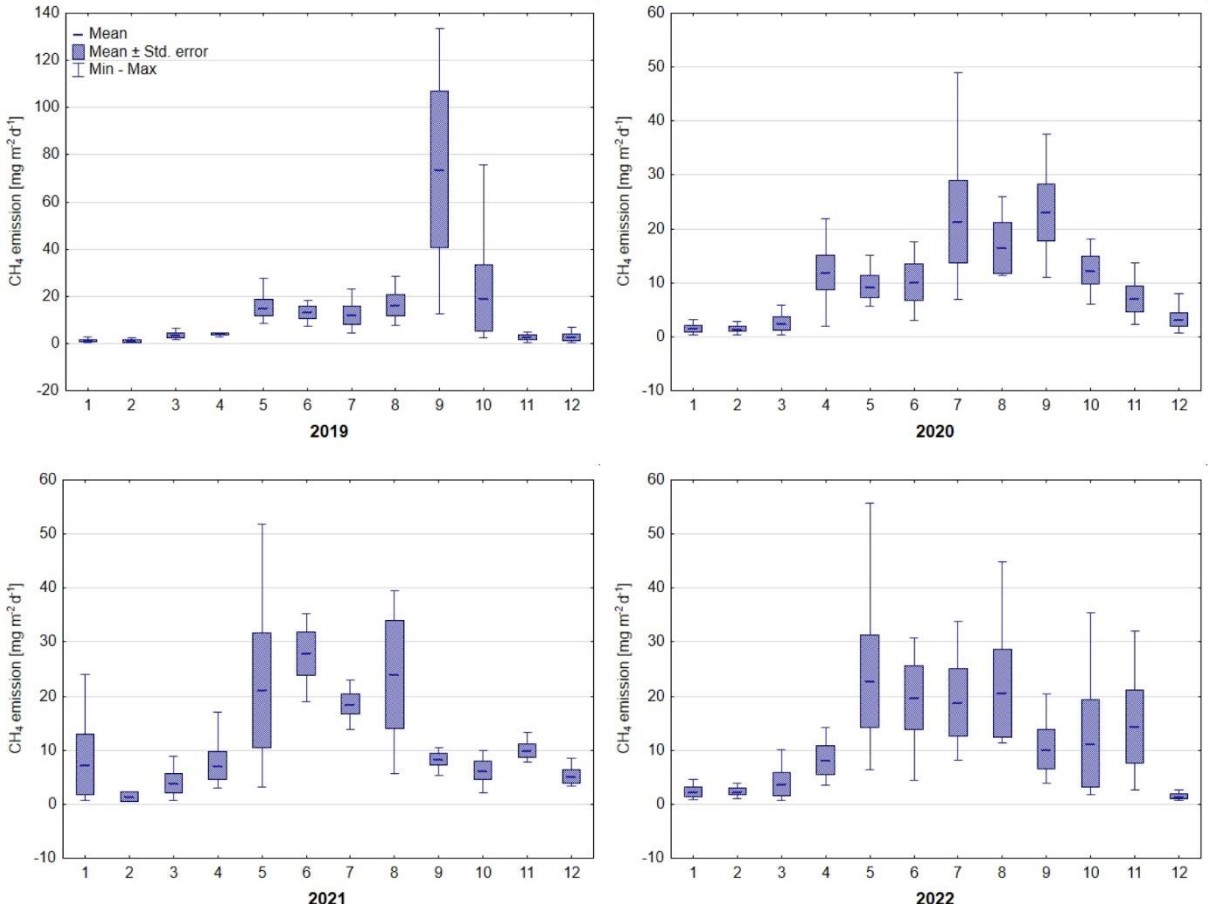

**Figure 4.** Variability of methane emissions from the surface of Lake Kortowskie in individual months, divided by years of study - average
values, standard error of the mean, and minimum and maximum extreme values.

Other studies indicate that oxidation in the surface layer affects 15-90% of methane entering from deeper layers during
autumnal circulation (Michmerhuizen et al., 1999; Schubert et al., 2012; Sanches et al., 2019). In Lake Kortowskie, the
hypolimnion (the layer >9 m deep) occupies only 9% of the lake volume, so the expected "storage" emissions should not be
of great significance. However, the lake also has a deoxygenated metalimnion (Fig. 1), which, together with the hypolimnion,
occupies 45% of the lake volume. Under particular weather conditions (strong wind, cooling), the epilimnion layer of the lake
can be additionally enriched with matter from the metalimnion, as observed in September 2019, when episodes of extremely
high methane emissions occurred in Lake Kortowskie.

The monthly results also highlight elevated methane emissions in autumn 2022 (Fig. 4), when both air and water
temperatures remained above 10°C until November 13, 2022 (Fig. 2). Similarly warm conditions were also observed in October
2019, when a high average emission and a large spread of recorded results were also noted.



### 3.2 Relationships between methane emissions and environmental factors

The relationships between methane emissions and meteorological and hydrochemical indicators revealed a series of associations, although they were quite complex and not always straightforward. Spearman correlation analysis, computed due to the non-normal distribution of $CH_4$ emission data (right-skewed distribution, skewness 4.37), demonstrated the existence of several statistically significant correlations, evident throughout the study period as well as in individual years and seasons (Table 3).

A very strong correlation was observed with temperature indicators, both air temperature and water temperature (statistically significant correlations at $p<0.001$). On one hand, this implies seasonality in methane emissions, as also evidenced in the data comparison in Figures 3 and 4. On the other hand, such a strong correlation may indicate the potential influence of climate warming on the magnitude of emissions from the studied lake. The correlation between $CH_4$ emissions and temperature was equally strong in each year of the study; however, it was not observed within individual seasons. Of particular interest is
the relationship between emissions and temperature in the summer season, whereby cooler periods were associated with higher emission values. Additionally, statistically significant correlation also occurred with wind speed, the occurrence of smaller temperature gradients (indicating better water mixing in the epilimnion), and in relation to poorer oxygen conditions in the epilimnion (Table 3). It is probable that periodic rinsing of the upper metalimnion layer occurred, delivering additional methane to the epilimnion, resulting in increased emissions. This is likely due to the sharp oxygen gradient (oxycline) in this lake layer
(Fig. 1). However, this remains a hypothesis that would require further investigation for confirmation. It appears to be intriguing, as it would imply that the release of some storage emissions also occurs during the summer stratification period, before the lake enters the autumn overturn phase.

    The recorded correlations also indicate the significant influence of water waving (surface water movement), which is only partially related to wind speed but also depends on wind gusts, wind direction, short-term changes in wind direction, and wind
duration. Based on available literature, it can be inferred that wind speed is typically considered in assessing methane emission magnitudes from lakes when calculated based on the current $CH_4$ concentration in the water, whereas water agitation on the surface (wave action) is not taken into account. However, it is worth noting that wave action exhibited higher correlation coefficients, yet both the average wind speed and wind gust magnitude showed positive values of r in all annual and seasonal configurations, though not always statistically significant.






**Table 3.** Spearman correlation coefficients for individual indicators relative to methane emissions from the surface of Lake Kortowskie, for the entire study period (n=198), in individual years (2019: n=51; 2020: n=52; 2021: n=46; 2022: n=49), and in seasons (winter: n=70, spring: n=38; summer: n=54; autumn: n=36); statistically significant values at the level of p≤0.05 are marked in red.

| Indicator | 2019-2022 | 2019 | 2020 | 2021 | 2022 | Winter | Spring | Summer | Autumn |
|---|---|---|---|---|---|---|---|---|---|
| Atmospheric indicators | | | | | | | | | |
| Atmospheric $CH_4$ concentration | -0,102 | -0,195 | -0,098 | -0,092 | -0,288 | -0,023 | 0,153 | 0,007 | -0,270 |
| 2-days mean air temperature | 0,685 | 0,688 | 0,699 | 0,653 | 0,713 | 0,134 | 0,306 | -0,227 | 0,209 |
| Air temperature during measurement | 0,644 | 0,676 | 0,658 | 0,626 | 0,646 | 0,046 | 0,112 | -0,320 | 0,138 |
| Dew point temp. | 0,551 | 0,576 | 0,529 | 0,575 | 0,550 | -0,141 | 0,270 | -0,274 | -0,030 |
| Cloud cover | -0,231 | -0,389 | -0,030 | -0,378 | -0,136 | -0,066 | 0,188 | 0,104 | 0,040 |
| Relative humidity | -0,344 | -0,444 | -0,312 | -0,293 | -0,286 | -0,180 | 0,102 | 0,039 | -0,191 |
| Wind speed | 0,230 | 0,092 | 0,277 | 0,437 | 0,167 | 0,334 | 0,288 | 0,430 | 0,228 |
| Max gust of wind | 0,249 | 0,150 | 0,357 | 0,395 | 0,275 | 0,169 | 0,408 | 0,333 | 0,313 |
| Air pressure | -0,003 | -0,045 | -0,133 | -0,091 | -0,048 | -0,227 | -0,069 | 0,153 | -0,090 |
| Water indicators | | | | | | | | | |
| Temperature | 0,736 | 0,796 | 0,741 | 0,684 | 0,716 | 0,427 | 0,473 | -0,011 | 0,255 |
| Temp. gradient in epilimnion | 0,220 | 0,101 | 0,249 | -0,005 | 0,352 | -0,139 | -0,126 | -0,300 | -0,140 |
| Oxygen saturation | 0,279 | 0,480 | 0,202 | 0,315 | 0,195 | -0,148 | -0,188 | -0,417 | -0,138 |
| Oxygen conc. | -0,356 | -0,391 | -0,385 | -0,312 | -0,384 | -0,199 | -0,345 | -0,461 | -0,237 |
| Electrolytic conductivity | -0,472 | -0,508 | -0,700 | -0,170 | -0,282 | -0,385 | 0,043 | 0,053 | -0,205 |
| pH | 0,287 | 0,416 | 0,319 | 0,314 | 0,015 | 0,219 | -0,021 | -0,123 | 0,020 |
| Redox potential | 0,008 | -0,359 | -0,367 | 0,328 | 0,498 | -0,203 | 0,318 | -0,023 | 0,126 |
| Chlorophyll-a | -0,073 | n/d | 0,301 | -0,203 | -0,098 | 0,125 | -0,525 | -0,060 | 0,046 |
| Turbidity | 0,324 | n/d | 0,595 | 0,230 | 0,220 | 0,259 | -0,501 | -0,031 | 0,201 |
| Secchi depth | -0,380 | -0,397 | -0,613 | -0,140 | -0,340 | -0,305 | 0,445 | 0,190 | 0,299 |
| Waves | 0,519 | 0,624 | 0,479 | 0,586 | 0,497 | 0,524 | 0,642 | 0,577 | 0,281 |

The results of correlation analysis were corroborated by the conducted PCA analysis in the system of factors: seasonality
(Factor 1, mainly based on thermal indicators), and physicochemical indicators (Factor 2) (Fig. 5). When using the full set of measurement data, the two main factors extracted explain only 43% of the variance, indicating that the influence of other factors is significant. Probably, short-term variability of conditions, including emerging weather anomalies, which affect the lake's functioning and disrupt the stability of the boundary between the epilimnion and metalimnion layers, as well as the influence of autumnal overturn bringing oxygen-depleted waters from deeper layers to surface waters, play a considerable role,
constituting a kind of limnological anomaly in itself.



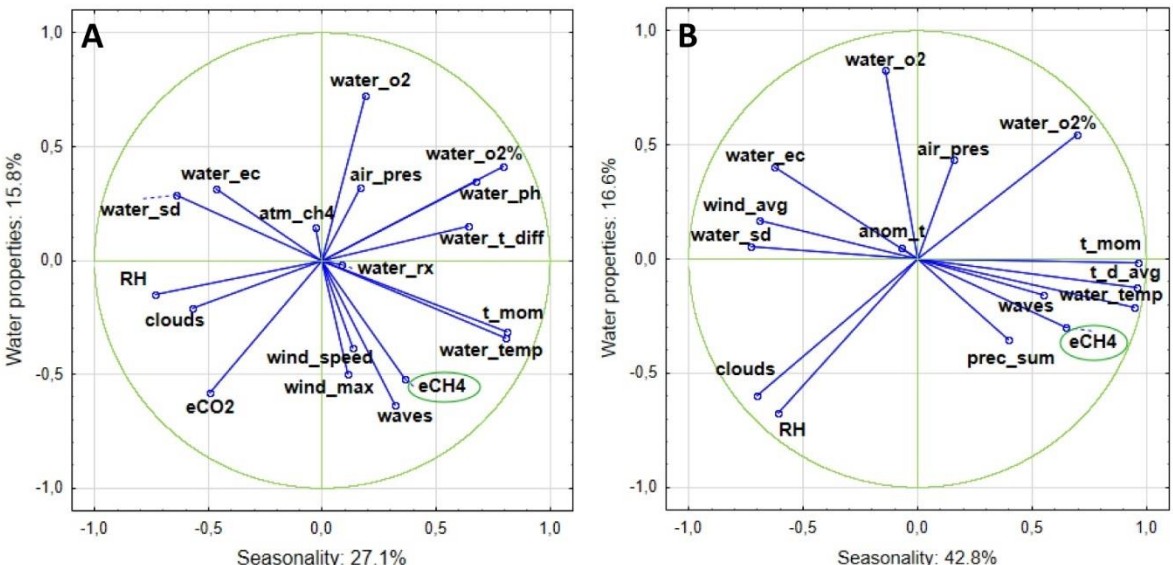

**Figure 5.** Results of PCA analysis for seasonality and physicochemical water indicators; A. analysis based on the full set of measurement data (n=198); B – analysis based on monthly mean data (n=48); % values on axis descriptions represent % of explained variance;

indicator abbreviations: eCH4 is methane emission from lake surface; eCO$_2$ is CO$_2$ emission (measured but not analyzed in this study); t_mom is air temperature at measurement moment; water_t_diff is difference in water temperature in the epilimnion; atm_ch4 is methane concentration in atmospheric air above the lake surface; water_ec is electrolytic conductivity; water_sd – Secchi depth; RH is relative humidity; clouds is cloud cover; wind_avg is monthly mean wind speed; prec_sum is monthly precipitation sum; t_d_avg is monthly mean temperature; anom_t is monthly temperature anomaly relative to the 1991-2020 average


The directional vectors of individual variables grouped together in the following manner: methane emission, according to this analysis, exhibited affinity with water and air temperature, wave action, wind speed, and, in monthly data, also with precipitation sum. Similar configuration of directional vectors in principal component analysis was obtained by Zhao et al. (2017) when analyzing methane emissions from a shallow lake in China. They observed a strong affinity of methane emissions

with water temperature and turbidity, and an opposite pole with electrolytic conductivity, pH, oxygen concentration, and redox potential.

In a separate analysis for each year, the extracted factors 1 and 2 explained over 60% of the variance, yet the directional vectors of individual variables differed across the years (Fig. 6). The year 2020 stood out from the others in terms of the spread of factor coordinates in the conducted analysis, and the directional vector of methane emission was more aligned along the

seasonality axis. Overall, 2020 was a very warm year (annual average temperature 9.5°C, Table 1), but it had an exceptionally warm winter and relatively cool summer. In the case of annual data, the directional vectors followed a repeatable pattern, with methane emission positioning itself in the same group of indicators with temperature, wave action, and precipitation, although precipitation in the dry year of 2022 clearly separated from the identified group of factors. However, the average wind speed




indicator appeared to deviate from the expected position on the graph plane. This indicates that the averaged wind speed value
significantly differs from the instantaneous values occurring during methane emission measurements.

**Figure 6.** Principal Component Analysis (PCA) results for monthly mean data, for each year of the study; Factor 1 primarily relates to seasonality, while Factor 2 relates to water quality indicators; analysis based on the dataset of monthly mean measurements (n=48); percentage values in the description of axes indicate the explained variance percentage; indicators are denoted as in Figure 5.



### 3.3 Regression models and the forecast of future methane emissions

Based on the obtained results of correlation analysis and principal component analysis (PCA), indicators most strongly associated with methane emission from the lake surface were identified, and multiple regression analysis was conducted to formulate equations describing the observed relationships. Water temperature is generally referred to as the primary factor determining methane production in lake ecosystems, thus influencing the emission magnitude to the atmosphere (Bastviken, 2009; Natchimuthu et al., 2014; Natchimuthu et al., 2015; Wik et al., 2016), so this indicator was first considered. For Lake Kortowskie, the regression model for methane emission ($eCH_4$), considering water temperature ($t_w$), takes the following form

($R^2 = 0.37$):

$$(1)\ eCH_4 = 0.97 + 0.80 \cdot t_w$$

Considering air temperature ($t_A$), the regression model takes the form of equation (2), with the value of $R^2 = 0.31$. Both

models have not very high coefficient of determination $R^2$, indicating that temperature is an important factor determining methane emission magnitude but only explains a limited part of the observed variability during the study period.

$$(2)\ eCH_4 = 2.89 + 0.80 \cdot t_A$$

A stepwise multiple regression model for methane emission, based on atmospheric factors, identified the following factors as statistically significant at the $p=0.05$ level: air temperature ($t_A$), wind speed ($ws$), and average cloud cover on the measurement day ($cl$). The model takes the form, with a coefficient of determination $R^2 = 0.39$:

$$(3)\ eCH_4 = -7.42 + 0.84 \cdot t_A + 1.87 \cdot ws + 0.62 \cdot cl$$


According to this model, under average parameter values in an average year, methane emission is 8.3 mg m$^{-2}$ d$^{-1}$, and for warm year conditions: 9.4 mg m$^{-2}$ d$^{-1}$. These values are significantly lower compared to those obtained in measurements. In the next step (4), a forward stepwise multiple regression model was applied, using all water-related indicators that showed statistically significant correlation with methane emission. As a result of the analysis, the following factors were included in

the model as statistically significant: water temperature ($t_w$), wave action ($wv$), and Secchi depth ($sd$). The model takes the following form in this configuration ($R^2 = 0.51$):



$$(4)\ eCH_4 = -15.9 + 0.94 \cdot t_w + 4.22 \cdot wv + 4.92 \cdot sd$$


This regression model has a much higher coefficient of determination $R^2$. For average conditions during the study period, model (4) yields a methane emission result of 11.8 mg m$^{-2}$ d$^{-1}$ (for the years 2019-2022: 11.9; 12.2; 11.8; and 11.3 mg m$^{-2}$ d$^{-1}$; seasonally for winter, spring, summer, and autumn: 4.6, 12.1, 20.6, and 14.9 mg m$^{-2}$ d$^{-1}$), which is very close to the values obtained in the conducted research. Model (4) provides a good reflection of the values obtained in measurements, however, it
is difficult to use it for forecasting future emission changes because while water temperature can be predicted, wave action and Secchi depth seem challenging to foresee. Therefore, another multiple regression model (5) was developed, which incorporates meteorological indicators available from long-term measurements. The analysis was conducted based on the monthly average values of the main weather indicators: monthly mean temperature ($t_{MA}$), monthly precipitation totals ($p_{MS}$), and mean relative humidity ($RH_{MA}$), whose long-term trends can be associated with climate change. These indicators showed statistically
significant correlation with methane emission at $p<0.05$. While temperature correlation is often reported as a factor influencing $CH_4$ emission, studies indicate that precipitation also affects emission levels (Schubert et al., 2012, Sanches et al., 2019). The regression equation (5) for Lake Kortowskie takes the following form ($R^2 = 0.66$):

$$(5)\ eCH_4 = -1.53 + 0.89 \cdot t_{MA} + 0.053 \cdot RH_{MA} - 0.003 \cdot p_{MS}$$


Using meteorological data for Olsztyn since 1951, the predicted values by model (5) were determined for the period preceding the conducted study, and forecasted values for these indicators for the year 2050 and 2100 were established based on the best fit of the extrapolated trend lines of temperature, precipitation, and relative humidity. These values roughly correspond to the regional climate simulation in 2050 and 2100 according to the CMIP6 model for scenarios ranging between
SSP2-4.5 and SSP3-7.0 according to the Copernicus Interactive Climate Atlas (atlas.climate.copernicus.eu/atlas).

Regression model (5) yields good results regarding methane emissions values for the entire study period and for annual values  (Table 4). However, in the seasonal context, it shows higher results for winter (though the absolute error is small), and noticeably underestimates autumn emissions. This may be due to storage emissions, triggered during the autumn circulation, the symptoms of which appeared in the obtained emission results, but their inclusion in the model is not possible with the
current dataset.






**Table 4.** Summary of the results of regression model (5) with comparison of data obtained from the model and from measurements, and forecast of emission size change until the year 2050 and 2100.

| Period | Input data | | | Results from regression model (5) [mg m⁻² d⁻¹] | Measured eCH₄ [mg m⁻² d⁻¹] | Difference between model and measurement [%] | Estimated change in eCH₄ [%] compared to: | |
|---|---|---|---|---|---|---|---|---|
| | $t_{MA}$ [°C] | $RH_{MA}$ [%] | $P_{MS}$ [mm] | | | | 1991-2020 | 2019-2022 |
| 2019-2022 | 9.0 | 77.7 | 52.2 | 10.4 | 11.72 | -10.9 | | |
| 2019 | 9.6 | 76.3 | 56.0 | 10.9 | 13.69 | -20.4 | | |
| 2020 | 9.5 | 77.8 | 55.6 | 10.9 | 10.11 | +7.7 | | |
| 2021 | 8.1 | 79.5 | 56.4 | 9.7 | 11.84 | -17.9 | | |
| 2022 | 8.9 | 77.4 | 40.6 | 10.4 | 11.57 | -10.3 | | |
| Winter 19-22 | 0.5 | 86.6 | 42.1 | 3.4 | 2.60 | +29.3 | | |
| Spring 19-22 | 7.4 | 67.7 | 45.8 | 8.5 | 9.43 | -9.8 | | |
| Summer 19-22 | 18.8 | 72.4 | 84.0 | 18.8 | 18.29 | +2.7 | | |
| Autumn 19-22 | 9.4 | 84.3 | 36.8 | 11.2 | 16.55 | -32.4 | | |
| 1951-1980 | 6.8 | 81.8 | 52.4 | 8.7 | | | | |
| 1991-2020 | 8.0 | 79.9 | 53.5 | 9.7 | | | | |
| 2050 forecast | 10.2 | 75.0 | 55.8 | 11.4 | | | +18 | +9 |
| 2100 forecast | 12.6 | 72.0 | 58.5 | 13.2 | | | +38 | +28 |

It is not yet clearly determined how methane emissions from lakes will change with the progression of global climate warming (Bastviken, 2009), which is understandable given the many uncertainties about the future of the climate system, particularly the difficulty in predicting which greenhouse gas emission reduction scenario the world will follow. It is known that potential methane production increases fourfold with a temperature rise of 10°C, and the optimum temperature for methanogenesis is significantly higher than the *in situ* temperatures currently present in aquatic ecosystems (Bastviken, 2009).

It is also anticipated that rising temperatures are one of the factors contributing to the eutrophication of waters (Meerhof et al., 2022), and the increase in aquatic ecosystem productivity is associated with a rise in pH values, which in turn favor methanogenesis. According to various studies, the magnitude of $CH_4$ emissions from lakes is positively correlated with indicators reflecting ecosystem productivity, such as total phosphorus and chlorophyll-a concentrations (Juutinen et al., 2009; Deemer et al., 2016; DelSontro et al., 2016; West et al., 2016). Model simulations (Beaulieu et al., 2019) suggest that increased

eutrophication (assuming a 30% rise in global lake productivity) could result in a 30-90% increase in methane emissions by 2100, primarily due to heightened overall anthropogenic pressure and only partly as a direct consequence of climate change. This increase in emissions could be one of the significant limitations in mitigating climate change to below 2°C relative to the 1850-1900 average (Collins et al., 2018). Additionally, studies by Davidson et al. (2015) suggest that the progression of eutrophication has a greater impact than the process of global warming itself. Literature indicates that as climate warming



progresses, the mitigation efforts related to reducing anthropogenic methane emissions will be offset by the increasing emission streams from natural sources, such as water bodies and wetlands (Beaulieu et al., 2019). On the other hand, better management of eutrophication, particularly by limiting external nutrient input, can significantly reduce methane emissions into the atmosphere.

Regression analyses conducted for Lake Kortowskie indicate that only changes in the main climate components, while 465 continuing the current trends, could cause an increase in methane emissions from the lake by over 30% by 2100 (Table 4). In all regression models (1-5), temperature plays the role of the main factor. In each of these models, the statistical significance level for temperature was p<0.001, and in each case, the b coefficient had a positive value. Therefore, the regression analysis leads to the conclusion that rising temperatures result in increased methane emissions from the studied lake surface.

## 4 Conclusions

Studies conducted on Lake Kortowskie have shown that methane emissions from the lake exhibit high temporal variability, but they display significant annual repeatability and distinct seasonal patterns. However, this is a complex process with high variability, influenced by both meteorological and limnological factors, including the seasonality of the lake itself, which is characterized as dimictic with spring and autumn periods of overturn and summer stratification.

Methane emissions recorded in Lake Kortowskie were relatively high, averaging 11.79 mg m$^{-2}$ d$^{-1}$, considering its geographical location. This may be associated with the advanced eutrophication of the lake, which is characterized by only moderate ecological status according to the European Union's Water Framework Directive. The highest emission values occurred during the summer (averaging 20.6 mg m$^{-1}$ d$^{-1}$), with high emission values also appearing in the autumn during the transition to the fall circulation phase. There were also events of increased emissions during the summer stratification period 480 (with a maximum of 134.4 mg m$^{-2}$ d$^{-1}$, and in a single measurement, 264.4 m$^{-2}$ d$^{-1}$), likely due to the transfer of methane from the upper metalimnion to the surface layer during cooling and periods of strong winds.

During the study period, annual average temperatures were recorded to be 2.2°C higher than the average temperature from 1951-1980, indicating a rapid progression of climate change in the region. Trends in the main climatological indicators suggest that this process is likely to continue. Regression analyses for methane emissions from Lake Kortowskie indicate that only 485 changes in the main climate components, assuming the current trajectory continues, could result in an increase in methane emissions from the lake by over 30% by 2100.






**Acknowledgements**

The results presented in this paper were obtained as part of a comprehensive study financed by the University of Warmia and
Mazury in Olsztyn, Faculty of Agriculture and Forestry, Department of Water Management and Climatology (grant No.
30.610.008–110).

Funded by the Minister of Science under „the Regional Initiative of Excellence Program".

**Data availability**

All raw data will be made available on request from the author.

**Competing interests**

The author has declared that there are no competing interests.

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
