# Peer review of "Seasonal and annual variability of methane emissions to the atmosphere from the surface of a eutrophic lake located in the temperate zone (Lake Kortowskie, Poland)"

_EGUsphere, 2024_

## Author Comment (AC1)

Response to Reviewer 1 Comments

First of all, I would like to thank you for the thorough analysis of my work and apologize for the delayed response; I was waiting for the second review to be received. I greatly appreciate all of your comments and apologize for the aspects I was unable to address due to the constraints under which I conducted the research presented in my paper. Below, I will address Reviewer 1 comments, with my responses marked in blue.

**general remarks**

This paper presents weekly (!) floating chamber measurements in a eutrophic lake over a period of 4 years – a quite impressive dataset. The author uses this data to calculate monthly and annual $CH_4$ emission from the lake. Using linear correlation analysis he develops an empirical model to predict $CH_4$ emissions from climate variables with a rather high power. That model is then used to predict future $CH_4$ emissions based on extrapolated historical climate data.

Although the topic of aquatic $CH_4$ emissions is subject to intensive research in the last years and decades it is still an actual topic with many unsolved questions. The rather high temporal resolution of this study offers a good potential to contribute to our knowledge about $CH_4$ dynamics in lakes. Unfortunately the manuscript does not really use that chance. There are a number of serious issues which need to be addressed before the paper can be accepted.

The paper lacks a clear research question or hypothesis. Just saying "We want to quantify $CH_4$ emissions from a particular lake" is not sufficient for an international journal.

I acknowledge that the research hypothesis is insufficiently emphasized in the paper; however, I believe the content of the article demonstrates that the assessment of emission levels was not the sole objective of my research. It was, however, a crucial aspect due to the lack of similar studies in the northeast region of Poland, which is characterized by a large number of lakes (the Masurian Lake District), typically exhibiting high levels of eutrophication. Furthermore, my literature review indicates that methane emission studies have not been conducted in this region to date. Another objective was to identify additional connections that are rarely considered in the literature, such as those related to water wave action and detailed meteorological data. In my study, these factors are analyzed over a period of four years of year-round observations. Therefore, I believe this aspect sets my research apart from other studies on methane emissions from lakes, many of which are conducted within a single year and often limited to the summer half-year period.

It is not very clear what new knowledge is provided. It is well known that temperature is a major regulator of $CH_4$ production in lakes and thus, that $CH_4$ emissions will increase with warming. There are numerous studies quantifying $CH_4$ emissions from particular lakes. The author need to make clear what makes his study outstanding. I thought a bit about this. In my eyes the strong points in this paper are:
- Weekly resolution
- 4 years covered
- Special lake with partly anoxic epilimnion
- Wave height measured

The author does not exploit these points. Instead of analysing weekly temporal dynamics he even aggregated the data to monthly resolution.

For the purposes of statistical analysis, the data were not aggregated to a monthly resolution; I worked with measurements from each individual sampling date. Monthly data were compiled solely to present them to the reader in a more concise form in the figure, and in the statistical analysis, where I relied on meteorological data that were available in monthly resolution. Therefore, at this stage, I adjusted my data to match that resolution. In general, all four of the mentioned points were considered in the analyzed data, though I acknowledge that they should have been addressed in more detail. My primary goal was to trace the annual and seasonal variability and capture the relationships between methane

emissions and other factors, as well as estimate the potential future changes in emissions. In fact, the entire paper focuses on this topic.

Of course, temperature is a well-known factor affecting methane emissions; however, my study provides insights, based on observed variability in emissions and environmental factors, as to the extent of potential increases in $CH_4$ emissions. In my opinion, this is not merely a confirmation of what is already well-known, but a conclusion derived from specific measurement data.

Your annual budgets are strikingly similar. Does that mean that short term fluctuations are not relevant on the annual scale (Morales-Pineda et al., 2014)?

Apparently, these fluctuations are not as significant, at least in the case of my study object. The research has shown that the observed high short-term variability is largely averaged out on an annual scale. I appreciate the Reviewer's attention to this aspect, as it may be a more important finding than I initially thought. I will take this into account when refining the discussion of the results.
The work cited by the Reviewer is not entirely relevant here, as it concerns $CO_2$ rather than $CH_4$. Southern Spain is a different geographic region with a different seasonal distribution, and it only includes data from a single summer period. Therefore, it seems difficult to expect significant similarities with the lake studied in my research. In my study, there was no statistically significant correlation between $CH_4$ and $CO_2$ emission values.

Weak parts of the paper are the treatment of ebullition and the lack of interesting additional data like weekly probe profiles or $CH_4$ concentration in the water. Although the author observed little evidence for ebullition in his measurements I am not convinced that ebullition can be ruled out as a process in this rather shallow lake.

Ebullition was difficult to quantify using the method available for the study, which is why I focused on diffusive emissions. As I mentioned in the paper, episodic ebullition was only observed in the shallower measurement locations, whereas no ebullition was detected at the deeper study sites, located near the boundary of the epilimnion. Hence, the suggestion that the ebullition process was of lesser significance in the studied lake. However, this is only a suggestion, and I expect that its inclusion in the paper is not inappropriate. I addressed this phenomenon to the extent that my observations allowed.

Therefore, in line with Reviewer #2's suggestion, I plan to add the word "diffusive" to the title.

A considerable part of the discussion is rather hypothetically without showing more data from the water column. He writes that he did probe measurements along with the chamber measurements. These data need to be shown! I recommend contour plots (heat maps) of the relevant parameters.

Since the paper focuses on methane emissions, I did not want to dedicate too much attention to other parameters, aside from their relationship with methane emissions. However, if this is important for the readers of the paper, I would be happy to expand on this aspect. I can also include the full range of research results in the form of graphs. I believe that a supplement would be a suitable place for this.

The reference list needs to be updated. The literature on the topic is highly dynamic and there are many more recent papers on the topic.

I fully agree and I promise to expand the reference list in the revised version of the paper.

**detailed remarks**
l.9: I would replace "Despite studies on methane emissions " by a statement underlining the relevance of lake $CH_4$ emissions.

l.17: Here a clear research question or hypothesis is missing

I agree that the study requires the formulation of a clear hypothesis. I propose Hypothesis 1: Weather patterns affect lake ecosystem functioning, which in turn influences both the intra-annual variability and cumulative methane emissions to the atmosphere. Hypothesis 2: Weather conditions and lake functioning allow for obtaining a measurement signal that enables estimation of potential changes in methane emissions in response to ongoing climate change. In preparing the revised version of the manuscript, I will further consider how to precisely formulate these hypotheses.

l.25: I guess water and air temperature were highly correlated. Maybe just write "temperature" without differentiating air and water.

In my data set, the air and water temperature had a correlation coefficient of 0.93, indicating a very high correlation indeed. Nevertheless, air temperature remains a distinct indicator from water temperature.

l.32: there should be a more recent reference than Dlugokencky. I would also remove Oh et al here.

Regarding atmospheric methane concentrations in the pre-industrial era, nearly all sources cite similar literature. I have not been able to find any recent studies that alter the 700 ppb value as the pre-industrial concentration. Are more recent publications truly necessary here?

l.47: There are more recent references on global $CH_4$ emissions from Lakes. For example (Rosentreter et al., 2021; Delsontro et al., 2018).
l.50: There is more recent literature than Bastviken 2004.

None of these papers are more recent than Johnson et al. (2022), which I referenced at the beginning of the paragraph. I cited the older works due to the interesting findings they contain. Thank you, however, for the additional recommendations; I will consider them when making revisions to the article.

l.55: Remove "current". You may also refer to thermal convection here.
l.60: Refer to Rosentreter et al.
l.67: "to attempt to assess" is not a good formulation here. Write which questions you want to answer.

Thank you for these comments. I agree with them.

l.71: Maybe replace CRDS by "a portable GHG analyser". Replace "flow chamber" by "floating chamber".

I would prefer to continue using the CRDS method, as that is the method I applied in my measurements. The term "portable GHG analyzer" generally refers to portable NDIR meters, which are a different and significantly less precise measurement method.

l.73: "… where to my knowledge $CH_4$ emissions from lakes were never published before"

Thank you, that would indeed be more precise.

l.78: Are these coordinates refer to the lake (then write so) or the site of measurement. Use past tense

These coordinates refer to the lake. Thanks for pointing that out.

l.90: I wonder if the lake treatment can be named "experiment"

In this specific case, the system installed on Lake Kortowskie can certainly be referred to in this way. From the beginning (since 1956), it has been regarded by its creators as a kind of experiment—various operational schemes were tested (different amounts of water discharged from the lake, various system operation schedules), and research was also conducted on the effects of the reclamation efforts. In Poland, this site is commonly referred to as the "Kortowski Experiment."

l.92: One profile is not enough to prove this statement.

This statement is based not only on my profile from Figure 1 but, as I further explained by citing Dunalska et al. (2007), on numerous observations from many years that indicate that the summer stratification in the studied lake is similar across different years. These are profiles from various years (I did not cite their source in my paper, as they come from a popular science book in Polish), which show that even in the 1950s, the lake's epilimnion reached a depth of 4–5 m, and the hypolimnion temperature was around 8°C. The atypical profile from 1989 resulted from the intensive drainage of the hypolimnion as part of the experimental lake restoration.

[Figure]

l.93-94: Why is this information relevant for the paper?

This is part of the description of the lake, explaining that the summer thermal stratification follows a stable pattern in different years, except during periods of intensive use of the hypolimnion water removal system (which was not in operation during my measurements). I do not understand why this information would pose any problem.

l.111: use past tense.

I agree, thank you.

Table 1: May be move to the supplement. Any rational why is humidity reported?

In my opinion, it is important to consider the climatic conditions under which the studies were conducted, especially in comparison with data from long-term reference periods. However, if the Editor deems the Supplement the appropriate place for this data, I will agree with that. Humidity is one of the

indicators in the table because I included this indicator in one of the regression models in Section 3.3. This indicator also changes in response to climate change in a less favorable direction (RH decreases, leading to increased evapotranspiration and worsened hydrological conditions, thus increasing drought risk). Therefore, I considered it important and included the data in the table.

Figure 2: Move to the supplement and add wind speed data.

As mentioned above, I can move the figure to the Supplement if the Editor decides so. Adding wind speed is a good idea, and I will make this change while revising the article.

l.136: I wonder whether there was a diurnal pattern of wind speed. Since all flux measurements were done during the day this could lead to an over-estimation of daily CH$_4$ emissions. Wind speed is often lower in the night.

This pattern does indeed occur quite frequently, but mainly in the summer. In other seasons, it is not necessarily the case. The midday hours, when I conducted my measurements, are usually not the peak wind speed hours but rather the average value for the daylight hours. During nighttime hours, the wind is often weaker (although not always). However, I am afraid that I do not have enough data to make corrections to my measurement results and replace them with simulated data. This is, however, a very interesting aspect that I plan to explore in my future research.

l.146: Is the equation correct? I cannot reproduce the unit mg/m2 d from it. Don't you need atmospheric pressure to convert ppm to mole? Please check the equation. You may also have a look at Unesco/Iha (2010).

To my knowledge, the formula is correct for the measurement method I used. I have checked this multiple times to avoid such a basic error in the calculation of results, which would be a disastrous situation. The unit mg/m$^2$ comes from taking into account the specific density of methane, which is in kg/m$^3$, considering the temperature. The calculations were performed for standard atmospheric pressure conditions. Due to the low variability of this indicator, I did not introduce corrections related to it. However, roughly, the variability of atmospheric pressure across the entire data set would have an impact on the results at about 0.1%.

l.160 This exclusion of 2 of 5 replicates is very unusual. I am sure such a procedure reduces data variability. But you cannot remove data just because they differ a bit.

In typical laboratory measurements and analyses, they are performed in 3 repetitions. I had a margin in this rule with 5 measurements, which is why I could afford such a preliminary data processing approach. It is worth noting that each of the 5 repetitions was already an averaged result from a continuous 3-minute measurement.
I performed this procedure for data analysis purposes in order to obtain the best possible averaged results, as the goal was to determine time-based (monthly, seasonal, annual) emission averages for the studied lake and to create the best possible models predicting emission changes in the future.

Reviewer #2 also pointed out this aspect, which is why, in the revised version of the paper, I intend to base my analysis on the average values of all 5 measurements.

l.167: I know that fluxes can vary on very short timescales depending on wind fluctuations. That's why I usually measure wind during each chamber measurement separately.
l-170: You need to explain how you scaled up your measurements at the 5 sites to the entire lake! You need to do this area weighted: The outmost point should be representative for a large part of the lake surface.

I did not scale the measurement results to cover the entire lake. All results are in mg per m², not for the entire lake. This is similar to any such studies—researchers measure only the part of the emission that enters the measurement chamber. No one places a dome over the entire lake. The same applies to the eddy covariance method—one cannot measure the entire atmospheric flux, these are necessarily only point measurements. I did, however, describe the observed variability depending on the location of the measurement station (lines 213–217).

I believe that actual measurement results are more valuable than those modified into values derived from simulations. I am of the opinion that the selection of research points in the adopted configuration adequately represents the emission from the lake surface. The results from individual stations were, on average, correlated with each other at r = 0.85.

l.176: I am not convinced.

I explained in the Methods section why I was unable to accurately capture the phenomenon of ebullition in my study. Reviewer #2 suggested adding "diffusive" emissions to the title, which I had already considered earlier, and I have a positive opinion about this solution.

l.183: Show the probe data.

The data from the probe is summarized in Table 2. However, if the request is confirmed by the Editor, I would be happy to prepare a presentation of the detailed data in the form of charts. I believe the Supplement would be an appropriate place for this.

Table 2: why ius ther no average pH and two pH values in the other columns. Ranges should be indicated by "-", not by "+".

In studies related to water, a range is usually used rather than average pH values, due to the structure of this indicator (logarithmic scale). Of course, I could perform a calculation by converting to the concentration of H+ ions, calculate the average, and then convert it back to the average pH value, but this is not typically the standard approach for handling such data.

l.217: "between individual years"
l.225: replace "result" by "meand flux"
l.233: Lake area should not be problem if area specific fluxes are reported.

Thank you for these corrections.

l.235: There should be more recent references – especially on eutrophic shallow lakes.

I agree with updating the references, but the term "shallow lakes" in the literature typically refers to polymictic lakes, meaning lakes in which thermal stratification does not form during the summer. The studied lake is dimictic, so such a reference is unlikely to be appropriate. Polymictic lakes, due to the absence of a hypolimnion (or in the case of highly eutrophic lakes, the lack of an anoxic zone in the deeper parts of the lake), function in a distinctly different way.

l.244: Remove "taken"
l.245: Remove "equilibrium state with methane present in the"
l.247: Would be interesting to see a plot of CH4 flux versus wind and/or wave height. The wave height measurements are interesting. Would also be interesting how wave height depended on wind speed.

Honestly, I did not expect that wave dynamics would turn out to be such an interesting aspect of my research. Of course, I can prepare a presentation of this relationship in the revised version of the paper.

l.255: The treatment of ice cover is not correct. During ice cover emissions are zero. If CH4 accumulates under the ice there should be higher emissions after ice off. Since you measured weekly you should have flux data very shortly after ice off.

Yes, I must admit that I had a serious issue with this and couldn't decide what to do with the winter measurements during the ice cover period. In fact, I conducted the measurements under such conditions out of pure scientific curiosity, not knowing what to expect. Ultimately, I kept these results in the dataset as "potential emissions" for this period. Replacing these results with "zero" values will not significantly alter the overall interpretation of the results, nor would it notably impact the calculated annual average methane emissions (since the values were mostly small and there were only n=14 winter measurements in the entire dataset). However, if the Editor agrees with the Reviewer's suggestion, I will change this and recalculate the results.

Unfortunately, I have not much winter data to date, which could have been useful for analyzing how the disappearance of ice cover might affect methane emissions from the lake. Especially since the ice cover period is significantly shortening due to climate warming. In my region, from 1950s. to 1990s, the ice cover lasted continuously from December 20th to March 15th, on average, while now it typically lasts only a few weeks, and in some years, there is almost no ice at all (this was the case in my study site during the winters of 2019/20 and 2021/22). If I have more such data in the future, I will try to prepare a study on this topic.

l.269: You say that weather affected $CH_4$ emission. Can you be a bit more specific here saying which weather had which effect?

Of course I can. I will describe this in more detail in a revised version of the article.

l.284: Add "or emissions after ice off".
Figure 4: The range is rather high in some months and hardly to explain by differences in wind speed. I would be suspicious that the very high flux measurements were affected by ebullition. If you have continuous ebullition of very small bubbles you do not see jumps in your chamber data and would interpret it as diffusion.

This is very interesting, but these are just assumptions that are not supported by my research. In my measurement method: a highly sensitive CRDS spectrometer and real-time measurement with sampling 72 times per minute, distinguishing bubbles from diffusion is easy and unquestionable. I am attaching a graph of a single measurement, which gave the highest recorded result on September 11, 2019. This was definitely not the effect of ebullition, but rather intense diffusion, which was not entirely uniform over time. Bubbles produce a sudden, almost instantaneous signal, in the form of a large spike in concentration in the measurement chamber.
The Y-axis represents CH4 concentration in the measurement chamber [ppm], and the X-axis represents the measurement time [s].

[Figure]

Here is an example of a measurement where ebullition appeared:

[Figure]

Figure 4: The boxes cover both between site and between week variability. I recommend that you present the data with weekly resolution. Than the variability can be interpreted as spatial variability and you can look at sub-monthly temporal pattern.

Below, I present a figure showing the entire set of measurements. I am not convinced that it provides a clearer picture of the annual variability of methane emissions than the monthly aggregation. However, if the Editor of my paper agrees with the Reviewer's opinion, I will include such an aggregation in the paper or in the Supplement.

I would like to emphasize once again: for the statistical analysis, I used all the results, and they were not averaged to monthly or seasonal values.

[Figure]

l.313: You can calculate activation energy and Q10 of your temperature dependence of the CH4 flux. This allows the quantitative comparison with temperature dependence in the literature.

l.327: This is very new and interesting. Exploit more. Maybe have a look at marine literature, where the effect of waves on gas exchange have been studied intensively.

Thank you for your comments. In the revised version of the paper I will develop both of these points.

Figure 5: Why do we need Figure 5b?

Figure 5B shows that similar relationships occur even after aggregating the data to monthly values, which partially eliminates the point-effect of daily data. Additionally, Fig. 5B includes precipitation, which could not be incorporated into the daily data.

l.364: I do not really see that.

Well, in my opinion it is clearly visible in the picture.

l.370: This can be checked by looking at weather station data.

I agree, but I wrote about it here because it came out of the "retroactive" analysis, which I think is an interesting observation.

Section 3.3: it is clear that air temperature and water temperature should be correlated and that water temperature has an effect on methanogenesis in the sediment – but air temperature not. Why did you correlate air temperature here?

I agree with the general principle, but I decided to use air temperature in the regression analysis because air temperature data are much more widely available than water temperature data. Therefore, I wanted to demonstrate that there is a relationship between methane emissions and air temperature, which may be useful for collective and review studies, for example, in estimating future methane emissions from lakes at a regional or global scale.

Maybe a better way to come up with a statistical model is runing mixed models using R and the use of something like the AIC to compare the performance of different models.

Thank you for the suggestion, but unfortunately, I have never worked with the AIC method when comparing models. I will look into this method and try to use it in the future.

l.455: You discuss eutrophication here but your data to not show a correlation with chlorophyll. Does that mean eutrophication is not in important regulator in your lake?

I do not intend to claim that eutrophication does not influence methane emissions from lakes. However, the studied lake during the research period was characterized by a stable condition, with no progress in eutrophication observed during this time. Indicators related to eutrophication, such as chlorophyll-a, turbidity, Secchi depth, and conductivity, did not show statistically significant differences between years. Therefore, searching for a link between eutrophication and $CH_4$ emissions would be unjustified in this case. In my study, eutrophication reflects the advanced trophic state of the lake, rather than a change in that state.

l.457: This is a strong statement. Can you support this by quantitative arguments?

The quantitative argument here is based on estimates suggesting a 30-90% increase in methane emissions from lakes by 2100, which I mentioned earlier in the text. In the revised version of the paper, I will try to find additional data on this topic, based on projected increases in emissions, atmospheric concentrations, and the radiative forcing values of methane.

l.471: "Methane emissions from lake Kortowskie exhibited …"
l.471-474: this is not new
l.477: Also the observed seasonality is not new.

These may not be particularly new conclusions in light of the existing literature, but they are new in relation to the studied lake and, considering the gap in research on methane emissions from lakes in my region, they are also new on a regional scale.

l.480-481: This is new.

Thank you.

**References**

DelSontro, T., Beaulieu, J. J., and Downing, J. A.: Greenhouse gas emissions from lakes and impoundments: Upscaling in the face of global change, Limnology and Oceanography Letters, 3, 64-75, doi:10.1002/lol2.10073, 2018.

Morales-Pineda, M., Cózar, A., Laiz, I., Úbeda, B., and Gálvez, J. A.: Daily, biweekly, and seasonal temporal scales of $pCO_2$ variability in two stratified Mediterranean reservoirs, J Geophys Res-Biogeo, 119, 509-520, Doi 10.1002/2013jg002317, 2014.

Rosentreter, J. A., Borges, A. V., Deemer, B. R., Holgerson, M. A., Liu, S., Song, C., Melack, J., Raymond, P. A., Duarte, C. M., Allen, G. H., Olefeldt, D., Poulter, B., Battin, T. I., and Eyre, B. D.: Half of global methane emissions come from highly variable aquatic ecosystem sources, Nature Geoscience, 14, 225-230, 10.1038/s41561-021-00715-2, 2021.

UNESCO/IHA, (IHA), T. I. H. A. (Ed.): GHG Measurement Guidelines for Freshwater Reservoirs, UNESCO, 138 pp., https://www.hydropower.org/publications/ghg-measurement-guidelines-for-freshwater-reservoirs, 2010.

Thank you for these references, I will use them when improving my paper.

---

## Author Comment (AC2)

Response to Reviewer #2 Comments

At the outset, I would like to thank Reviewer #2 for conducting the review and thoroughly analyzing my preprint. I consider the Reviewer's comments to be valuable, and below I will attempt to address them. My responses are highlighted in blue font.

The paper *"Seasonal and annual variability of methane emissions to the atmosphere from the surface of a eutrophic lake located in the temperate zone (Lake Kortowskie, Poland)"* by Skwierawski is looking into the methane emissions from Lake Kortowskie for four years at five different stations, measuring emissions almost weekly. Skwierawski analyses the methane emissions in relation to a variety of environmental factors on all the data (n = 198) and on monthly average emissions using Spearman correlation, principal component analysis and linear modeling. Lastly the author uses the measured values to predict methane emissions from the lake in 2050 and 2100.

The author has collected an interesting dataset with a long time series of data. I am concerned about the placement of the chambers, as all chambers are placed within 1.5–3.5 meters of water depth, which leads the author to conclude negligible amounts of methane ebullition take place. Nonetheless, many papers are starting to conclude that ebullition occurs in the deeper parts of the lake and accounts for large fractions of the total methane emission with increasing emissions. Moreover, the analysis conducted in the paper needs a thorough check, as simple things such as intercorrelation were never considered. Additionally, I find that the extrapolation of methane emissions to 2050 and 2100 should be associated with much uncertainty. It has become a discipline in the papers on methane emissions to try and extrapolate further and wider, more often than not too much. The manuscript is also in need of more references on some of the important statements.

Overall, I find that the paper has a good-fair scientific significance due to the large amount of data collected. However, the scientific quality of the paper is fair. The statistics are missing from the manuscript, making it hard to conclude on the significance statements. Furthermore, I cannot find any information on how the data has been handled beforehand (scaled, intercorrelation, etc.). The presentation quality of the paper is fair; the language is easily understood, yet on several occasions, the reason for using analysis or the results is not revealed until later.

Thank you for this summary, and I mostly agree with the Reviewer's opinions. I will address the specific aspects in more detail later in my response, with regard to the particular comments and questions.

**In general**
Add statistical information when referring to correlations.
&
L133. Please indicate the method used to assess the correlation and statistical information (df, F-value, etc.).

These indicators (df, F-value) refer to the analysis of variance. The statistical indicators for correlation are r, p, and n, which I included in the presented results related to correlations. However, I will take note of this when preparing the revised version of the manuscript and will supplement the information where it is missing.

Title should reflect only diffusive emissions are considered.

I agree, I have considered this version of the title before and ultimately I agree that it would be the right solution.

**Abstract**
L 14. Please refrain from using abbreviations in the abstract.
L 24. "The studies" should be changed to "The results"

Thank you, I will make these changes.

**Introduction**
L36. Needs a reference
L37. Needs a reference

Indeed, I treated these statements as generally known information, but they require references. I will cite Collins et al. 2020 (10.1088/1748-9326/ab6039) and IPCC 2021 (10.1017/9781009157896) in this context, while also specifying the data: the methane lifetime is 11.8 years, and the global warming potential over 100 years (GWP100) is 27.0.

L45-46. I see your point, but it's counterintuitive to first state that lakes play a significant role and then state that the magnitude is difficult to estimate.

In my opinion, there is no contradiction in this statement: lakes play a significant role in the methane budget, as supported by both measurements and estimates. However, accurately estimating this role on a global scale is difficult to achieve.

L70. Belongs in the method.

I agree, but I wanted to emphasize here that the measurements were conducted using a relatively rare method, and I wanted to showcase this to the reader right from the beginning of the paper.

Figure 1. I don't understand the partitioning into partial catchments.

This is the classification commonly used in hydrology, distinguishing between direct and indirect catchments. The map shows that the southern part of the Kortowskie Lake catchment drains directly into the lake (the lake being the first receptor of water in the drainage streams), while in the northern part, the first receptor is the large Ukiel Lake. This is an important aspect of the lake's characteristics.

**Methods**
L130. What qualifies as a faulty measurement?

Occasionally, the device used experienced issues with smooth operation, and due to lag in the readings, it was not possible to extract a complete, uninterrupted measurement over the full 3-minute period. Fortunately, this occurred very rarely, mostly during extremely hot days with high humidity. Typically, it affected 1 out of the 5 measurements taken.

L132. The representativeness, please spell out what the measurements are representative off.

At this stage, I just wanted to verify whether my data were obtained under conditions that were representative of the weather conditions during the whole study period. And indeed, they were. The results would not have been representative if the conditions during the measurement periods had deviated from the average values observed over the study period.

L145. Why do you only use C0 and C180, when you have 216 measurements (72*3) of methane increase? You could use the linear increase.

To determine the change of concentration during the measurement period, I used the difference between the average at the beginning and the end of the measurement, as the increase in concentration was not necessarily linear throughout the measurement. Sometimes, during the

measurement, the slope of the curve changed. This occurred especially under more turbulent and rapidly changing weather conditions, which occasionally happened during measurements. The C0 and C180 results average this, typically small, but nonetheless existing variability.

L160-167. I am not convinced that this is a good method. Methane emissions are variable in space and time, which you concluded in the beginning. By removing these observations, you remove some of the noise, but the high emissions may be due to ebullitive emissions and thus very much as relevant as the low fluxes.

This approach was intended to reduce the standard deviation between individual measurements in order to obtain a more averaged mean value for the sampling date, for the purpose of processing the data over broader time intervals. However, upon reflection (and also considering the opinion of Reviewer #1), I agree that, from the perspective of interpreting temporal data, it was not the best approach. I will make this adjustment in the revised version of the paper.

L167-170. I believe that the sensor measures the changes in temperature and humidity within the chamber, so it's possible to determine the change in environmental conditions.

Unfortunately, the CRDS spectrometer with the measurement chamber that I used, in addition to measuring $CH_4$ and $CO_2$ concentrations, only measures absolute humidity, but unfortunately does not measure temperature.

L173. I would like you to point out in the title that you are only measuring diffusive methane emissions and not ebullitive.

Yes, I agree with that statement.

L137. As all measurements were done within 1.5–3.5 meters of water depth, you are likely to have low ebullitive emissions due to oxygen reaching the sediment, more wind disturbance and less accumulation of organic material. There might in fact be high ebullition in the deeper parts of the lake. Or in the reed belt where there is also high accumulation of organic material.

This is possible, but unfortunately I do not have data to confirm or reject such a hypothesis. However, the observations I made show that ebullition occurred only at shallower measurement sites, while at the sites furthest towards the water column, I did not observe this phenomenon even once. Similar observations for lakes are presented by DelSontro et al. 2016 (doi.org/10.1002/lno.10335), where ebullition was not observed at depths > 3 m. See Fig. 2 from their paper:

[Figure]

L182. Wrong parenthesis at NTU
Table 2. Use – rather than ÷ to display range. Units should be the same as the methods text even though they are equivalent.

I agree. Thank you.

**Results and discussion**
L216. missing mg after 14.6
L220. Also missing mg after 14.9
L224. Remove "prepared"

I will fix it. Thank you for your perceptivity.

L224-230. Remove the absolute values. As you are comparing it to your values, which are in unit area, it is irrelevant here what the total area emissions are.

The cited authors provided their data in terms of total area emissions, whereas I converted them to unit emissions based on their data. Therefore, I retained the original values from the referenced papers. If this is unnecessary, I can remove it.

L260. In my opinion, you should discard ice-covered periods, as you will have a buildup of methane underneath the ice, which will eventually be released when the ice breaks, however, by only measuring in a few locations by drilling, you will cause few areas where the methane can actually escape and thus elevated emissions here. This scenario might be the case in the instance with emissions of 24 mg m-2 d-1.

As I mentioned to Reviewer #1, I had a significant issue with how to handle the winter measurements during the ice cover period. In fact, I conducted these measurements out of scientific curiosity, and ultimately, I included these results in the dataset, treating them as 'potential emissions' for that period. Replacing these results with 'zero' values would not change the overall interpretation of the results and would only slightly affect the calculated annual average methane emissions (since n is only 14), but it also would alter the results of the individual calculated statistics in some extent. It's a shame because this was the most challenging part of the field measurements, but I understand the reservations about these results.

I would greatly appreciate a suggestion on how to handle these winter values in statistical calculations: should I leave them with zero values, or should I exclude them from the statistical analysis? Unfortunately, I find it difficult to make a decision on this matter.

L269. To my knowledge, a lot of people have tried explaining this variation, without luck. To state that the variability is attributed to weather conditions should at least be backed up further than just referring to a figure.

Thank you for this challenge, I will try to explain the observed variations in more detail in a revised version of the paper.

L307. Please explain prior how this skewness is calculated and what it means.

Skewness is a simple statistic that indicates whether the data distribution is close to normal or deviates from it. In this case, the distribution was positively skewed (positive distribution), meaning it had an "overrepresentation" of low numerical values. Skewness was calculated as one of the basic statistics in

the analyzed dataset as part of the statistical review before proceeding with further stages of data processing.

L310. Were tests made to look for intercorrelation between parameters? I would expect a high collinearity between water and air temperature.

Yes, I conducted such tests at the preliminary stage of data processing. The water temperature showed a strong correlation with air temperature (r = 0.93).

L320–322. Please rephrase, it's very hard to read this sentence.

I propose changing this phrase to:
"The conducted observations may indicate that the release of some storage emissions also occurs during the summer stagnation period, before the lake enters the autumn overturn phase."

L323–325. Need reference.
L326. Need references.

This generally refers to all papers in which methane emissions were calculated based on water concentration using the 'gas transfer velocity' equations of Liss and Slater (1974) and their subsequent modifications. There are many papers employing this method. This method does not account for wave action, but only considers the wind speed at 10 meters above the lake surface. Of course, I will add the appropriate references to support these statements.

L328. Did you calculate wave action, and if so, how? This is the first time we hear about it. It needs to be in the method section as well.

I measured the wave height directly at the measurement site. This is described in the methodology, see lines 187-191.

Table 3. I am certain many of the indicators would show collinearity which should be taken into account. What emission values are used here, is it average from each sampling period?

The table shows the correlations between methane emissions and meteorological and water indicators. I believe that the correlations between all the indicators will not be useful here. The results are based on the emission values from each measurement within the given period under consideration (years, seasons).

Section 3.2. This section is missing more discussion on the parameters, which group together with methane emissions, and why it is expected or not. What effect do the different parameters have on methane emissions? Right now, it is not used for much, or at least you do not say why you do it. I can read in the next section why you do it, but I would want to know beforehand.

I agree, in the revised version I will attempt to conduct a deeper exploration of this data resource.

L381. Yes, water temperature affects methane production, but that is different from methane emission. Water temperature also affects methane oxidation.

Of course, that is true, however, in my research, I was dealing with a black box type model and I only observed the effect of these opposing processes, in the form of $CH_4$ emissions into the atmosphere. This is one of the limitations of the study that I cannot avoid. From a practical point of view (i.e., in

terms of the final result – the impact on atmospheric concentration, and subsequently on the radiative balance), this final effect is probably the most important.

L411. "much" higher is pushing it here.

Well, I agree. However, in such inseparable conditions in which the lake operates (with numerous measured as well as unmeasured factors), such an R value seemed to me to be 'much' higher.

Table 4. Please indicate the confidence interval around the predicted values.

That's a good idea, thank you for this suggestion. In the revised version of the manuscript, I will recalculate the model, taking into account the limit values of the individual components and provide the confidence interval.

L449. It should also be mentioned that the methane oxidation will increase with temperature.

That's true, however, this was not the focus of my research. I can only comment on the final effect of these processes, which was the subject of my measurements.

L452. Add reference.

Both sentences, starting and ending on this line, are supported by references.

L467. B coefficient? I guess it is the slope.

The coefficient b is the regression coefficient (or more precisely, the unstandardized regression coefficient). 'Slope' is another term, I believe both are correct.

Section 3.3. Please indicate when values or trends from references are done *in-situ* or in laboratory experiments.

I admit that I do not understand this claim. Of course, most studies on methane emissions are based on field research (although laboratory experiments can likely be found). Perhaps the reviewer is referring to the fact that I emphasize that my research is direct in situ? If so, I only wanted to highlight with such statements that the entire measurement process was conducted in real-time and on-site, whereas most studies rely on collecting methane samples and later analyzing them in the laboratory. I don't see any reason to elaborate on this in the results section.

L480. Unit is wrong 264.4 m-2 d-1.

Of course this is a mistake, thank you.

**Data availability**
To my understanding Biogeosciences requires data to be uploaded in an online repository, which I encourage.

According to the journal's guidelines, there is no such requirement. However, if my paper will be accepted for publication and the editor recommends that I can publish the raw data, and I will agree to it without any reservations.